# Traditional Fabric and Medicinal Use Are the Leading Factors of In Situ Conservation of *Gossypium barbadense* in Central Brazil

**Kálita Cristina Moreira Cardoso** [1], **Guilherme Hoffmann Barroso** [2], **Fabio Oliveira Freitas** [3], **Ivandilson Pessoa Pinto de Menezes** [4], **Catarina Fernandes Silva** [5], **Nair Helena Castro Arriel** [6], **Valdinei Sofiatti** [6] and **Lucia Vieira Hoffmann** [6,*]

1   Post-Graduation Program in Conservation of Cerrado Natural Resources, Instituto Federal Goiano, Campus Urutaí, Urutaí 75790-000, GO, Brazil
2   Department of Economics, Administration and Sociology, Escola Superior de Agricultura Luiz de Queiroz, University of São Paulo, Piracicaba 13418-900, SP, Brazil
3   Embrapa Recursos Genéticos e Biotecnologia, Brasília 70770-917, DF, Brazil
4   Biology Department, Instituto Federal do Ceará, Campus Acopiara, Acopiara 63560-000, CE, Brazil
5   Escola de Agronomia, Universidade Federal de Goiás, Goiânia 74690-900, GO, Brazil
6   Embrapa Algodão, Campina Grande 58428-095, PB, Brazil
*   Correspondence: lucia.hoffmann@embrapa.br; Tel.: +55-6235332235

**Abstract:** The Sea Island cotton *Gossypium barbadense* has been present in Brazil for at least 750 years. Cultivated worldwide, the fibres present superior quality; therefore, farmers' seeds are an important genetic resource and in situ maintenance is essential to complement ex situ conservation. To understand how the species has been conserved in situ and investigate the socio-economic aspects which may ensure the continuity of its conservation, we conducted expeditions to three different municipalities situated in Brazilian Cerrado, Goiás state, Brazil—one of which is a traditional community, the quilombo Kalunga community—interviewed plant maintainers and compared our results with data from the Brazilian Institute of Geography and Statistics. There is hand spinning and hand weaving for home uses and commercialization within and outside the traditional community, which contribute to the continuity of the in situ conservation of *Gossypium barbadense*. Medicinal use is more determinant than hand weaving in deciding to keep plants and seeds. Fabric handicraft is a predominantly female, low-income activity. Interviews with cotton hand spinners indicated that in situ maintenance may be favoured by access to weaving looms and improvement in the marketing and sale of their products. Policies valuing handicrafts can ensure the continuity of biodiversity and disseminate and vivify traditions in addition to maintaining an income for the artisans.

**Keywords:** genetic resources; in situ maintenance; medicinal plants; Pima cotton; quilombo; Sea Island cotton

## 1. Introduction

*Gossypium barbadense* L. is native to the northern part of South America and extends into Mesoamerica and the Caribbean. It is also known as Pima, Long Staple, Sea Island, Egyptian, or Tanguis cotton [1]. Archeological evidence shows that cotton cultivation and use in South America go back at least 7800 years, with the earliest known variety being *G. barbadense* [2]. In Brazil, an archeological *G. barbadense* boll was found in Central Brazil (Januaria cave, Minas Gerais) dating back 750 years as estimated by radiocarbon dating [3].

*G. barbadense* is of commercial interest as its fibers are of superior quality compared with the main cultivated cotton, *G. hirsutum* L. [4], as well as having other qualities such as potential for medicinal use [5,6].

Hand spinning is practiced in some places [7] including traditional communities in Central Brazil [8]. It consists of ginning the cotton, carding the threads, and spinning,

sometimes using a wheel. Although it derives from different traditions, there is a certain universality in the practice [7].

As is the case with many other tropical countries such as Haiti, Brazil saw the formation of several hinterland communities by formerly enslaved people who eluded captivity [9]. In Brazil, these communities came to be known by the term *Quilombos*. *Kalunga* is one of these communities. It is distributed in sections of three municipalities of the Central Brazil *Veadeiros* Plateau, with a National Park and a Conservation Unit. Members of the *quilombo* have recently enjoyed the possibility of increased income from tourism, mainly from selling various handcrafted items or local produce in various small storefronts.

Social and environmental changes have introduced strong threats to the in situ conservation of germplasm, and the tools to describe, measure, and encourage its maintenance must be developed. Considering the hypothesis that it is possible to characterize the social and economic factors leading to in situ conservation, the objective of the expeditions and interviews was to relate *G. barbadense* maintenance to community sustainability. Results indicate simple actions or policies which may foster in situ maintenance. The present paper seeks to further the understanding of the pool of cotton biodiversity that is maintained by several communities in Brazil that grow small amounts of cotton for ornamental and medicinal purposes, as well as for fabric production.

## 2. Materials and Methods

In order to obtain information pertaining to the in situ conservation of heirloom cotton species such as *G. barbadense*, we conducted expeditions to three municipalities of interest in Central Brazil. The three municipalities present a tropical climate and are in the Cerrado (Savanah) Biome, except for Santana do Araguaia, which is in a transition from Cerrado to Amazon.

The first was Guaraíta, state of Goiás, on 27 November, 2021. The second was Cavalcante, where members of the *Kalunga* community, the biggest *Quilombo* in Brazil, were interviewed. The expedition lasted five days in July 2022. The other was *Santana do Araguaia*, in the northernmost region of the country, during four days in August 2022. Four additional fabric artisans were contacted as a comparison: one in northeast Brazil, two in São Paulo, and one in the same central region, but from another municipality (Santo Antonio de Goiás).

During these expeditions, we contacted members of the local community and obtained their consents (in the case of the *Kalunga* community, we also obtained permission from the *Kalunga* Association, as per Brazilian law 13.123 (16 November 2015) and Decree 8772 (11 May 2016)). In the *Kalunga* community, thread spinners were located and contacted by a woman from the community, and all hand spinners known by her—possibly all from her community—were interviewed, except two who were traveling. At Santana do Araguaia, we located small rural properties using the map and received help from a local guide. The region of small farms covers more than half of the municipality, with almost all the properties being land reform settlements. We drove along a dirt road through approximately a third of the municipality. We managed to see the cotton plants and requested information from the residents regarding who grew other plants; only one plant owner did not want to be interviewed. At Guaraíta, all plant owners who currently hand spin, and did not refuse to be visited, were interviewed.

At localities, a questionnaire—available in the Supplementary Materials—was applied to the plant owners about their use of cotton, in the hope that these data would be useful to identify economic and social opportunities to further the in situ conservation of cotton biodiversity.

Whenever there were cotton plants being grown by these individuals, their characteristics were recorded in order to allow for species identification. This was carried out to identify current levels of cotton biodiversity in these communities. Seeds were collected for ex situ maintenance.

Socio-economic data about Brazilian hand weaving were obtained from the National Survey by Household Sample (PNAD), a publication of data about the Brazilian workforce published by the Brazilian government through the Brazilian Institute of Geography and Statistics (IBGE).

It should be noted that in 2016 the PNAD's methodology of data collection underwent several changes. The data published from the very first issue of the PNAD up until 2015 are referred to as "traditional PNAD"; data published since then are referred to as "continuous PNAD". Most importantly for our purposes, since 2016 the category pertaining to artisanal weave workers has been removed and merged with leatherworkers. With this constraint in mind, we will use and present data from 2015 and 2021 in our study.

## 3. Results

Of 36 respondents, 33 reported to grow cotton from heirloom seeds acquired from neighbors or family. Nineteen of them reported using cotton for hand spinning the fiber as well as for medicinal use, seventeen used cotton for the medicinal properties of the seeds only, and only one of the women grew cotton for mere ornamental purposes. Those who hand-spun the cotton did so for one of three purposes: 5 of them did so for crafting items used by themselves, 7 for selling the wires directly, and 7 either crafted items they later sold or had someone else craft the items and sold those (Figure 1).

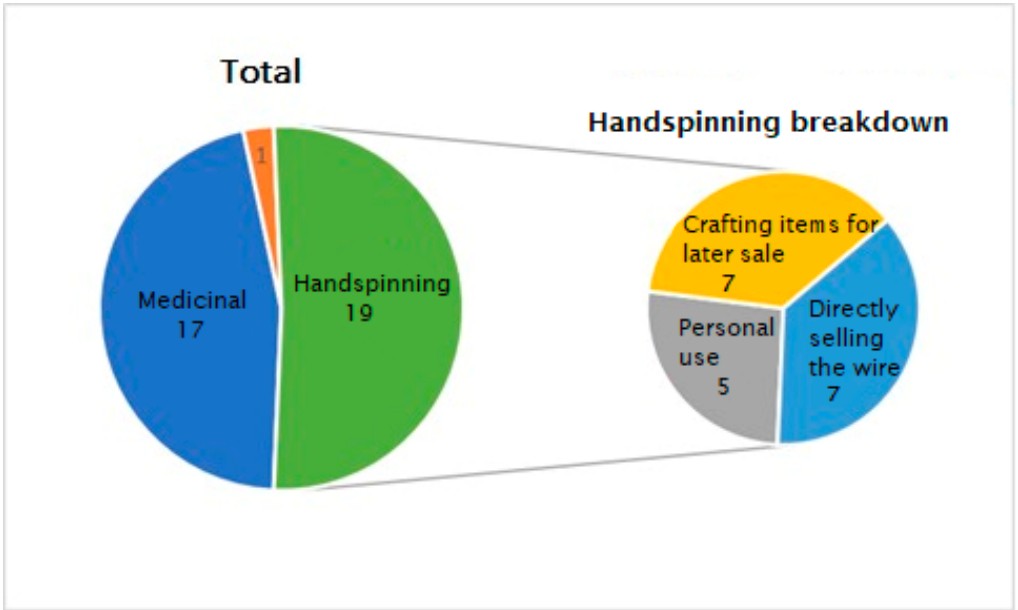

**Figure 1.** Two pie charts with information regarding cotton use by the respondents. The pie chart under "fabric artisan's breakdown" shows the use given to the manual spinning of cotton wires by the artisans. On the left, the number one represents the single plant used as an ornamental plant.

### 3.1. Guaraíta and Itapuranga

Five artisans were interviewed in Guaraíta and Itapuranga. Four of them reported to be in possession of cotton plants; they spin, but do not weave. The wires obtained are frequently used in manual weaving looms. While manual ginners, spindle, and wheel are simple and it is possible to spin without them, it is not possible to weave without looms, which are generally expensive, large, and dependent on maintenance. Spinning and weaving require training to be practiced, and in this municipalities as well as in Cavalcante, there are hand spinners who cannot use weaving looms simply because it is difficult to find people to repair them when they are broken. What makes the Guaraíta region unique are the traditional meetings of the people practicing cotton manual-spinning. We participated in one of these meetings (October 2021) and it was a festive event where people from the community work together, as well as an opportunity to meet people and

strengthen social ties. The map showing the places where plants were found is shown in Figure 2 (georeferentiation is shown in the Supplementary Materials). The average earning in Guaraíta from this activity among the five artisans was BRL 225 per year, corresponding to only 15% of the median earning of an occupied person in Brazil in 2021. For most of them (4/5), however, this is not the main source of income.

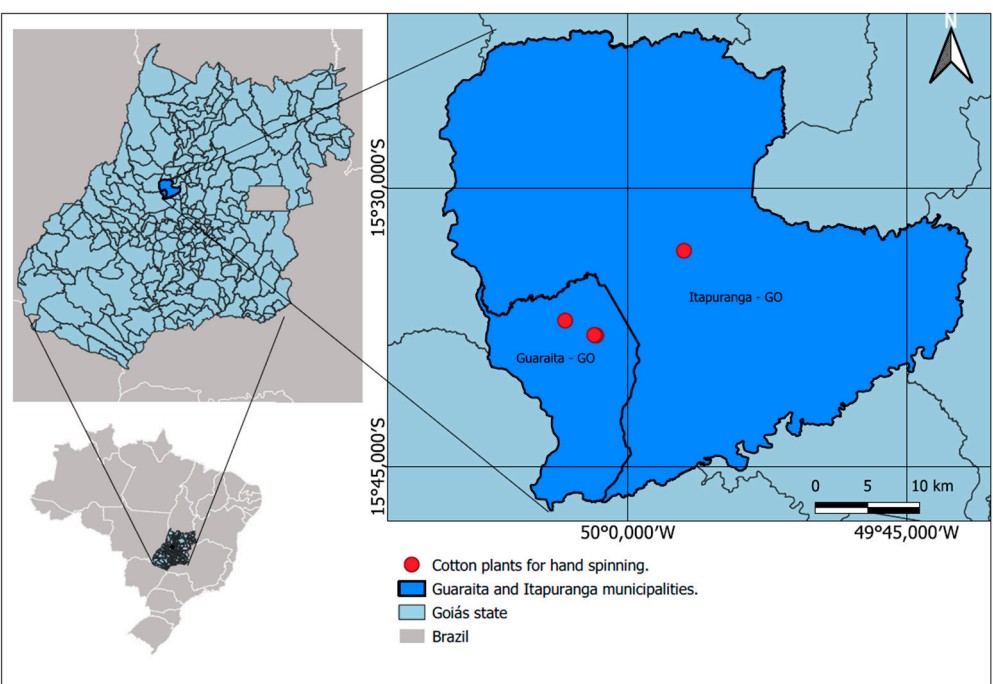

**Figure 2.** *Gossypium barbadense* cultivated by hand spinners who accepted to be interviewed in Guaraíta and Itapuranga.

### 3.2. Kalunga Community

In the community, 14 women reported to be in possession of cotton plants, grown near their house in a rural area. Only one of them used it solely for the medicinal properties of the cotton plant leaves; all the others reported to also use cotton for manual spinning.

Critically, most of the interviewees (12/14) reported that they trade cotton yarn and craftsmanship as a source of income. The main handicraft products are rugs and bags. The greatest difficulty in selling the products is transportation from the rural community to the village, since the roads are dirt; furthermore, there are rivers, but no bridges. Access is only possible with 4 × 4 traction cars that enable crossing of the rivers. None of the spinners interviewed had their own vehicle. There is a store in Cavalcante village where this handicraft is sold, and there are sales in the Alto Paraíso de Goiás municipality. The commerce seems to be growing, and two weaves have recently been acquired by a store in the village of Cavalcante. All the spinners also use cotton for medicinal purposes. The map of the collections in Cavalcante is in Figure 3. Georeferentiation is shown in the Supplementary Materials.

Almost all of the respondents were over 40, with three exceptions: a woman of 31, a girl of 18, and a girl of 12, showing that the practice is being learned by the young. A boxplot displaying the age of the hand spinners who answered the questionnaire from the four municipalities can be seen in Figure 4:

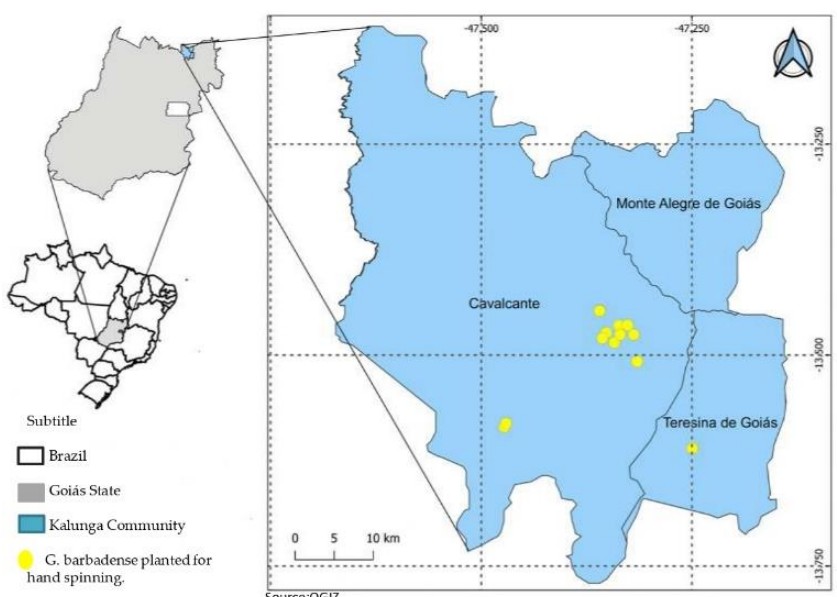

**Figure 3.** *Gossypium barbadense* cultivated by hand spinners in the Cavalcante and Teresina de Goiás municipalities. All hand spinners accepted to be interviewed and gave seeds to be preserved at Embrapa germplasm bank.

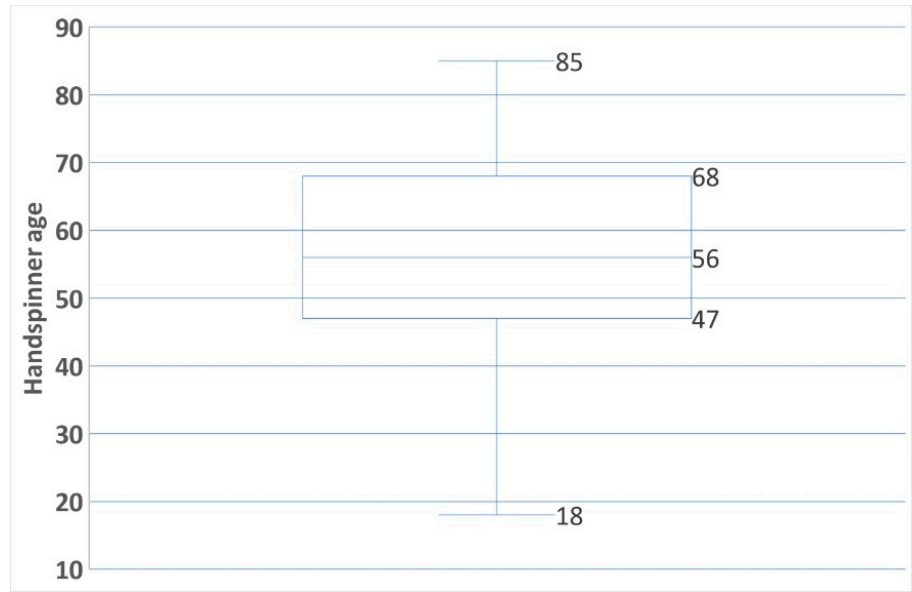

**Figure 4.** Boxplot of the age of hand spinners, showing that the practice is being learned by the young.

The average earning with craftsmanship in the *Kalunga* community was BRL 189.89 per year.

Four out of fourteen plants presented purple leaves, bracts, and stems. Thirteen of them presented tightly united seeds, known as kidney seeds. Leaves were always big.

### 3.3. Municipality of Santana Do Araguaia

In contrast to the *Kalunga* community, the population of this municipality mostly consists of farmers who occupy small allotments of land, doled out as part of the land reform program undertaken by the Brazilian government, as shown on the database of the Ministry of Agriculture, Livestock and Food Supply (MAPA) [10]. The points of collection are presented in Figure 5. Georeferentiation is shown in the Supplementary Materials.

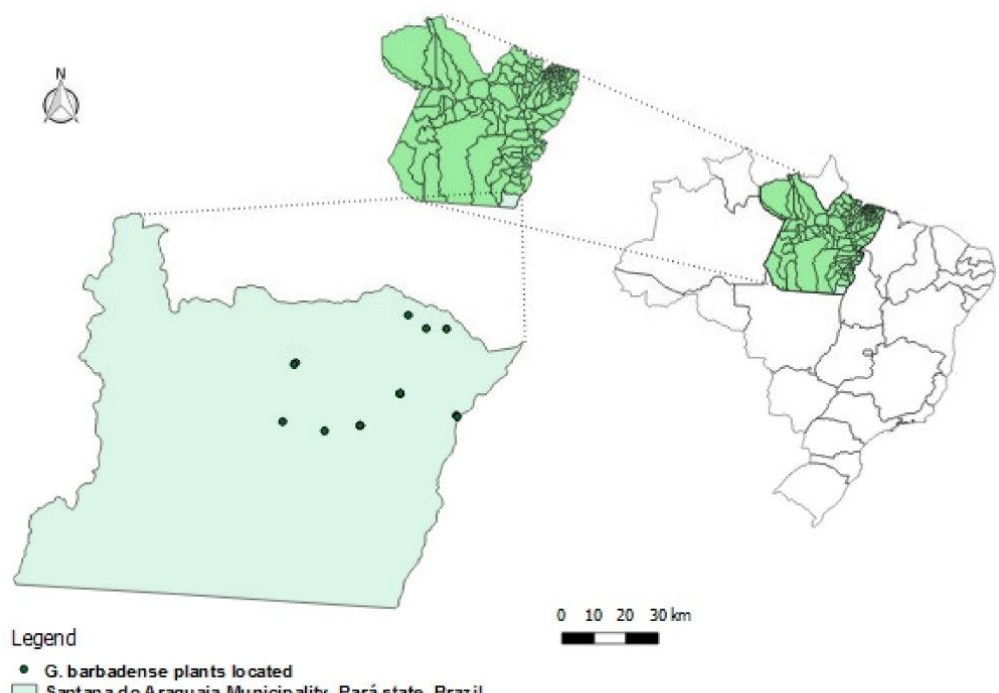

**Figure 5.** *Gossypium barbadense* cultivated for medicinal purposes localized at Santana do Araguaia.

None of the respondents in this particular community reported growing cotton for its fiber. Among the people interviewed here, the unanimous reason for the cultivation of cotton was extraction of cotton seed oil for medicinal purposes.

Three out of fourteen plants collected at Santana do Araguaia presented reddish leaves, bracts, and stems. Ten bore kidney seeds; the other four presented free, non-united seeds. Leaves of all plants were big, as is typical of the *G. barbadense* species.

### 3.4. Socio-economic Data from the National Survey by Household Sample

As a separate category for fabric artisans is inexistent in the latest version of the PNAD, we will henceforth use traditional PNAD data, which, having started in the 1970s, was discontinued after 2015. In continuous PNAD, which started in 2012 and continues to be used, occupation is classified differently, and the separate category is nonexistent (there is, however, a broader category for artisans "of fabrics, leather and similar materials").

In the 2002–2015 period, there was a marked and continuous decrease in the number of fabric artisans, starting from 172,702 workers in 2002 and finishing with 42,619 workers in 2015. During the same period, the occupied population of the country grew from 66.5 million to 82 million.

There was a continuous increase in average years of schooling of fabric artisans; however, this average is still considerably lower for fabric artisans than for the general population. In 2002, the schooling of fabric artisans lasted 4.8 years on average, and 6.7 in 2015. In the whole Brazilian population, schooling lasted 7.4 and 9.6 years on average in 2002 and 2015, respectively. The proportion of women among fabric artisans was always high, with 92.5 of women in 2002 and 82.9 in 2014, which was the lowest proportion recorded overall. Across all occupations, the proportion of woman grew gradually from 2002 (37.9) to 2015 (41.0).

The age of workers shows a tendency toward growth, considering both all workers as well as fabric artisans only. We made this comparison to investigate whether there are young workers working as fabric artisans, and whether young people are learning the traditional practices. Considering that the average age is a random variable, we can then model these data as

$$y(t) = \alpha + \beta t$$

where $y$ is the age in any given year, $t$ is the year, and $\alpha$ and $\beta$ are parameters of the population. Using the least square method to estimate $\alpha$ and $\beta$ yields, the increase in all occupied people was $\beta = 1.99$, while when considering fabric artisans only the increase was $\beta = 0.131$.

### 3.5. Medicinal Use

The main part of the plant used for medicinal purposes is the leaf but soaked seeds may also be used. Only once, among the Kalungas, was the use of external fruit parts or roots reported. In the three communities visited, the infusion of the leaves was used as an antimicrobial and anti-inflammatory agent, in the external treatment of wounds, and in postpartum baths. In the Kalunga community, it is used to treat respiratory viruses.

## 4. Discussion

*Gossypium barbadense* is a genetic resource valued for its fiber quality, which is especially long [11]. It is grown as a cash crop in several countries worldwide [12–14]. In the US, Upland and Pima, cotton account for around 95.5% and 4.5%, respectively, of total fiber production [13]. In Brazil, it is almost not cultivated commercially since cultivars are not being developed. It may be used by industry as well as by local craftsmanship [15]. It also has medicinal uses thanks to its antimicrobial and healing properties [5,6]. This species crosses with the main cropped cotton, *G. hirsutum*, domesticated in Mexico [16]. It has been found in the wild in several places in South America, where it originated [15], although it does not occur naturally in every country. In Brazil, it occurs only when grown in backyards and gardens (https://www.cnpa.embrapa.br/albrana/ (accessed on 24 February 2023)). Its conservation is, therefore, inextricably linked to its cultivation.

Interview results show hand spinning is linked with conservation in the state of Goiás, but not in Pará. Previous Embrapa research shows that about 80% of plants in the northern region—in Acre, Amapá e Roraima [17], Pará [18], and the Amazonas states [19]—are grown only for medicinal purposes. In the Tocantins state, some of the cotton was used for hand spinning [17,20]. The Brazilian book published in 1890, titled *Popular Medicine*, in which plants' medicinal uses are described, states regarding the Gossypium genus that "*The flowers, leaves and seeds of the cotton plant are emollient and are used in Brazil in an infusion prepared with 4 g of flowers or leaves and 360 g of boiling water, for lung irritations and dysentery. In Pernambuco state (Northeast Brazil) they use the seeds for difficult menstruations. The roots are diuretic*". The *Gossypium* species that was most distributed in Brazil during the 19th century must have been *G. barbadense*, while *G. hirsutum* var *marie galante* was also planted in northeastern states [21].

The growing access of the general population to textiles and medication, as well as the modification of rural landscapes in Brazil from small individual plots to sprawling mechanized plantations, seem to be responsible for the loss of continuity in the conservation of species, as well as in the knowledge of cultivation and use.

The Convention on Biological Diversity (CBD) [22], as well as the International Treaty on Plant Genetic Resources for Food and Agriculture [23], define in situ conservation as "conservation of the ecosystems and natural habitats and, in case of cultivated or domesticated species, where they developed their properties or characteristics". This should also be executed for *G. barbadense.* Public policy in support of biodiversity and gender equality can also support fabric craftsmanship.

Manual spinning activity is, therefore, more concentrated within traditional communities than outside them [24] and may be associated with fiber quality. Within traditional communities, some young inhabitants are learning traditional practices.

The purple color of plants, more common in the Amazonia region [19], were strongly present in the *Kalunga* community, suggesting ancient distribution in the region. The

acquisition of weaves and cars for transportation is the limiting issue. the characterization of local art and stores, as well as internet commercialization, will come naturally.

In situ and ex situ maintenance may favor cotton breeding for long fibers, which can be strongly favored by modern genetic technologies [11]. The main foreseen benefit of *G. barbadense*'s genetic introgression under commercial cultivation is the fiber quality, but phenolic compounds that may confer resistance to insects were also reported [25].

## 5. Conclusions

Although the relevance of *G. barbadense* can be differentially described by traditional people and enterprises dealing with commercial agriculture, fiber quality may be important for both. Genetic diversity is conserved by traditional agricultures who are facing economic changes and discontinuing agricultural practices. In situ conservation may bear more diversity than germplasm banks, therefore their maintenance may be monitored or even favored by governments. *G. barbadense* is conserved in the Brazilian Cerrado and Amazon by hand spinners, and incentives for weaving looms repair and acquisition and handicraft commercialization may foster cotton maintenance and community sustainability. *G. barbadense* is also maintained for medicinal use of the plant, mainly the leaves. Policies seeking to support handicraft could ensure the continuation of biodiversity and the continuation of tradition, as well as secure an income for the women that work in this area.

**Supplementary Materials:** The following supporting information can be downloaded at: https://www.mdpi.com/article/10.3390/su15054552/s1, Table S1. *Gossypium barbadense* plants georeferencing points, Kalunga communities. Table S2. *Gossypium barbadense* plants georeferencing points, Santana do Araguaia.

**Author Contributions:** Conceptualization: K.C.M.C., I.P.P.d.M. and L.V.H.; methodology: G.H.B., F.O.F., K.C.M.C., I.P.P.d.M., L.V.H. and V.S.; software, G.H.B. and C.F.S. Writing: all authors. Funding acquisition: L.V.H., K.C.M.C. and N.H.C.A. All authors have read and agreed to the published version of the manuscript.

**Funding:** This research was funded by Embrapa, Convention of Biological Diversity-CDB (Bio-Bridge initiative project coordinated by Food and Agriculture Organization-FAO) and Agreement for technical and financial cooperation between Embrapa and Pará Association of Cotton Producers.

**Institutional Review Board Statement:** The study was conducted in accordance with the Declaration of Helsinki, and approved by the Ethics Committee of Instituto Federal Goiano (protocol code 66971122.7.0000.0036, at 5 December 2022.

**Informed Consent Statement:** Ethic Committee: '*Comitê de Ética em Pesquisa CEP/IF Goiano*'. CAAE: 66971122.7.0000.0036.

**Data Availability Statement:** Not applicable.

**Acknowledgments:** To Ziany Neiva Brandão for the Santana do Araguaia map.

**Conflicts of Interest:** The authors declare no conflict of interest.

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
