# Peer review of "Traditional Fabric and Medicinal Use Are the Leading Factors of In Situ Conservation of Gossypium barbadense in Central Brazil"

_sustainability, doi:10.3390/su15054552_

Round 1

Reviewer 1 Report

This is a nice exploration essay (what they called "expeditions"), unfortunately, I can only see very little Science in it. The species, i.e., Gossypium barbadense,  in the surveyed sites, may be necessary for the sustainability that benefits the local residents, however, the effective conservation methods, the usage, and the functions, especially the medical functions are not well studied in the current manuscript.

I cannot give more comments on this manuscript unless the study on Gossypium barbadense is well conducted using a scientific perspective.

Author Response

Question Reviewer 1: This is a nice exploration essay (what they called "expeditions"), unfortunately, I can only see very little Science in it. The species, i.e., Gossypium barbadense,  in the surveyed sites, may be necessary for the sustainability that benefits the local residents, however, the effective conservation methods, the usage, and the functions, especially the medical functions are not well studied in the current manuscript.

I cannot give more comments on this manuscript unless the study on Gossypium barbadense is well conducted using a scientific perspective.

Answer: Thank you for your observation. We recognize we have had problems to put the hypothesis and methodology clearly. We have the suggestions from six reviewers and, therefore, we hope to have been able to review the description of the scientific question and development of our “Communication” over a methodology to study the sustainability on the conservation of a plant species in local economies. Genetic resources can be conserved in situ; the people who usually kept plant diversity, by planting it, are otherwise facing economical changes and discontinuing agricultural practices, which maintain diversity. We want to contribute to develop methodologies that may be used to monitoring in situ maintenance and verify the social and economic sustainability of it. The main modification on the text to show this is the last paragraph of introduction. Other small modifications have been made as well.

The relevance of Gossypium barbadense can be differentially seen by the side of the traditional people who conserves its diversity and enterprises dealing with commercial agriculture. We tried to briefly describe traditional people view in the introduction, and plant breeders and biotechnologists remains summarized at the first paragraph of the discussion.

The species is, like many others of the local biodiversity, necessary for the sustainability of the community. Seeds have been collected for ex situ maintenance, which is well stablished.

The scientific demonstration of the plant properties that lead to medicinal use are not well described in the literature, as far as we know, and is not object of this communication. The Brazilian book published in 1890, titled ‘Popular Medicine’, in which plants medicinal uses are described, says about the Gossypium genus that “The flowers, leaves and seeds of the cotton plant are emollient and are used in Brazil in an infusion prepared with 4 grams of flowers or leaves and 360 grams of boiling water, for lung irritations and dysentery. In Pernambuco state (Northeast Brazil) they use the seeds for difficult menstruations. The root is diuretic.’ The Gossypium species that was more distributed in Brazil during the nineteen century must have been G. barbadense, while G. hirsutum var marie galante was also planted at Northeast states (https://doi.org/10.1590/1984-70332015v15n1a4).

Reviewer 2 Report

Comments to the manuscript by Cardoso et al.

The manuscript "Gossypium barbadense preserved in situ in Central Brazil by fabric artisans and for medicinal use" is good because it documents the importance of wild plant species “Gossypium barbadense L.” used in Brazil's economic activities. It was written in a well-simplified way, and according to the complex results, it has resulted from surveys and reliance on some historical information.

Main observations:

(1) The language is acceptable.

(2) In the first mention of the plant species in the text, write it with the full name "Gossypium barbadense L.".

(3) It will be better to add a paragraph for study sites, including a brief description of the location, climate conditions, and others.

(4) The medicinal uses of the plant need to be expressed more and shown in detail, for example, diseases that the plant species is used to treat.

(5) Rethink about "key words"; the authors can add more frequent words in the manuscript.

Author Response

Question Reviewer 2: In the first mention of the plant species in the text, write it with the full name "Gossypium barbadense L.".

Answer:The following information was added to the first paragraph of methodology: The three municipalities present a tropical climate, and the biome is called Cerrado (Savanah), Santana do Araguaia being already a transition to the Amazon

Question Reviewer 2: The medicinal uses of the plant need to be expressed more and shown in detail, for example, diseases that the plant species is used to treat.

Answer: It has been added in a new item in results section (3.5)The main parts of the plant used for medicinal proposes is the leaf, but soaked seeds also may be used. Only once, at the Kalungas, the use of external fruit parts or roots were reported. In the three communities visited, the infusion of the leaves was used as an antimicrobial and anti-inflammatory agent, in the external treatment of wounds, and in postpartum baths. In Kalunga community, it is used to treat respiratory viruses.

Question Reviewer 2: Rethink about "key words"; the authors can add more frequent words in the manuscript.

Thank you for the observation, errors were corrected.

Reviewer 3 Report

The manuscript (sustainability-2147866) entitled “Gossypium barbadense preserved in situ in Central Brazil by fabric artisans and for medicinal use shows interesting data. However, after reading the complete text, I found that there are many problems, which can be summarized:

1)     The introduction section is just a grab of information from different sources, and only basic descriptive statements are given. I don’t think they are appealing to readers because they already know well about that. Revise your introduction section more logically or scientifically. Why is this communication important to publish?

2)      Provide a questionnaire as supplementary material.

3)      The logic of the article is confusing and not clear.

4)      Results lack the mechanistic approach.

5)      Rewrite your conclusion based on your main findings of the study.  

Author Response

Question reviewer 3: The introduction section is just a grab of information from different sources, and only basic descriptive statements are given. I don’t think they are appealing to readers because they already know well about that. Revise your introduction section more logically or scientifically. Why is this communication important to publish?

Answer: Thank you for the comment and for pointing out this problem. The first and last paragraphs of the review have been modified. In the first one we put information about the species and, in the last one, we add the hypothesis, formulated methodology and intended use of the results.  We observed there is a lack of scientific approach to prospect sustainability aspects that drives population to the continuity of in situ genetic resources preservation. The reason why we thought that it should be published is that because it may be a source of information and methodology help do design appropriate approaches to prospect on in which measure economical aspects or costumes drive plant diversity maintenance, and which measures can be taken to support it. This is fundamental because we know that diversity outside germplasm banks are greater than in them, at least for most of the tropical countries.

Question reviewer 3: Provide a questionnaire as supplementary material.

Answer:It was added. Thank you for the suggestion.

Question reviewer 3: The logic of the article is confusing and not clear.

Answer: We apologize for that, and we hope it became more clear now, with the many modifications made in the original manuscript. We are thankful for the observations.

Question reviewer 3: Results lack the mechanistic approach.

Answer: Modifications have been made. Please note that we opted for communication instead of scientific article because it is a general perception survey approach

 Question reviewer 3: Rewrite your conclusion based on your main findings of the study.  

Answer: Modifications have been made.

Reviewer 4 Report

Revision required

Author Response

We recognize we have had problems to put the hypothesis and methodology clearly. We have the suggestions from six reviewers and, therefore, we hope to have been able to review the description of the scientific question and development of our “Communication” over a methodology to study the sustainability on the conservation of a plant species in local economies. 

Reviewer 5 Report

Dear Authors

Title: Gossypium barbadense preserved in situ in Central Brazil by fabric artisans and for medicinal use

Author(s): Cardoso et al. (2023);                             MS#: Sustainability_2147866

This is an interesting communication and the authors tried to explore the leading factors resulting in situ conservation of Gossypium barbadanse in central Brazil. The authors also investigated some of major hindering factors as well using interviews/questionnaires method. However, I would like to suggest a few minor amendments which are as follows;

1.      Title need rephrasing to better depict the study. For example, “Traditional fabric and medicinal use are the leading factors of in situ conservation of Gossypium barbadanse in Central Brazil”

2.      Line 13: It is not clear that authors referring to which cotton species here. Please add full common name say “Sea Island Cotton” if it’s Gossypium barbadanse. Secondly, the native area of this species is Colombia, Ecuador and Peru, and an introduced species for all other geographical places. Therefore, domestication and being native of an area are two different aspects. Please revise accordingly

3.      I would like to recommend use of conservation/conserved (which is more equivalent to a sustainable use) term than preservation/preserved (=save by not using) in the MS.

4.      Please elaborate your study hypothesis, objective and potential outcomes in the last para of your introduction section.

5.      Please provide some socio-economic-cultural attributes of your 36 study participants in tabular form in M&M section.

6.      How the study participants were selected in the area? Please add

7.      Figure 1, extended plot text is not readable.

8.      You did not quoted any particular medicinal use/ recipe, or dosage of the species parts etc. Cotton seed oil is used to cure which ailments?

9.      Based on results and discussion presented in the MS, conclusion statement need some amendment to make it matching with title of study.

Author Response

Question reviewer 5: Title need rephrasing to better depict the study. For example, “Traditional fabric and medicinal use are the leading factors of in situ conservation of Gossypium barbadanse in Central Brazil”

Answer: The tile has been changed. Thanks for the excellent suggestion.

Question reviewer 5: 

Line 13: It is not clear that authors referring to which cotton species here. Please add full common name say “Sea Island Cotton” if it’s Gossypium barbadanse. Secondly, the native area of this species is Colombia, Ecuador and Peru, and an introduced species for all other geographical places. Therefore, domestication and being native of an area are two different aspects. Please revise accordingly

Answer: The common name Sea Island cotton has being added at the first line. The first lines is introduction has been changed, with a more adequate reference for the origin and common names of the Sea Island cotton. 

Question reviewer 5:  I would like to recommend use of conservation/conserved (which is more equivalent to a sustainable use) term than preservation/preserved (=save by not using) in the MS.

The modifications were made along the text

Question reviewer 5:  Please elaborate your study hypothesis, objective and potential outcomes in the last para of your introduction section.

Answer: Thank you a lot for this excellent suggestion. It was added: ‘Considering the hypothesis that it is possible to characterize social and economic aspects that lead to the continuity of germplasm maintenance through in situ conservation, the objectives of the expeditions and interviews were to relate G. barbadense maintenance to community sustainability. Results are used to indicate simple actions or policies, which may foster in situ maintenance. ‘

Question reviewer 5:  Please provide some socio-economic-cultural attributes of your 36 study participants in tabular form in M&M section.

Answer: As this was our first visit to the communities (Kalunga and Santana do Araguaia), we did not fill free to ask many questions, asking only about how much each participant earn with cotton handcraft activities, and trying to guess how our research can bring results in return for the interviews.

Question reviewer 5: How the study participants were selected in the area? Please add

Answer: The information was added at the third paragraph of the section Materials and Methods.

Question reviewer 5: Figure 1, extended plot text is not readable.

Answer: The images have been duly altered in accordance with the reviewer's suggestions

Question reviewer 5: You did not quoted any particular medicinal use/ recipe, or dosage of the species parts etc. Cotton seed oil is used to cure which ailments?

Answer:  It has been added in a new item in results section (3.5)The main parts of the plant used for medicinal proposes is the leaf, but soaked seeds also may be used. Only once, at the Kalungas, the use of external fruit parts or roots were reported. In the three communities visited, the infusion of the leaves was used as an antimicrobial and anti-inflammatory agent, in the external treatment of wounds, and in postpartum baths. In Kalunga community, it is used to treat respiratory viruses.

Question reviewer 5:  Based on results and discussion presented in the MS, conclusion statement need some amendment to make it matching with title of study.

Answer: Modifications have been made.

Reviewer 6 Report

The article is well written and the material is presented coherently and clearly. There are only a few minor suggestions to improve the article and to remove existing inaccuracies. 

1. Line 16. I suggest that the name of the province or other region of the country should be inserted in the abstract alongside the name of the place, so that the geographical location is immediately clear.

2.  Line 30. The subdivision 1.1. of the Introduction has no justification and I suggest that it be removed.

3. Line 33. The author of the taxon should be indicated: Gossypium barbadense L.

4. Line 36. The reference to the author of the taxon should be corrected: G. hirsutum L. (missing full stop after L and space).

5.  Lines 94-95. The sentence can be deleted and the reference to the figure (Figure 1) can be inserted in the sentence preceding it. 

6. In Figure 1, the colours of the numerals need to be adjusted. The numerals in the graph are now difficult to see because of the lack of contrast. I suggest that the numerals be in black. 

7. Figure 2. In the caption, descriptions of all parts of the figure should be added and the country, province or other geographic attributes should be indicated.  A mere reference in the legend is not sufficient.  The same applies to Figure 3.

8. Figure 4. No indication of what is on the Y-axis. It is necessary to add an entry. 

9. Figure 5. Figure caption must be updated (see note 7). The entries in the figure that are not place names should be translated into English.

10. Line 232. Remove unnecessary text.

11. Line 285. The reference should not be underlined. 

Author Response

Question reviewer 6:  I suggest that the name of the province or other region of the country should be inserted in the abstract alongside the name of the place, so that the geographical location is immediately clear.

Answer:  Very good suggestion, added.

Question reviewer 6:  Line 30. The subdivision 1.1. of the Introduction has no justification and I suggest that it be removed.

Answer:  Thank you for the observation, it was corrected.

Question reviewer 6:  Line 33. The author of the taxon should be indicated: Gossypium barbadense L.

Answer:  Added.

Question reviewer 6:  The reference to the author of the taxon should be corrected: G. hirsutum L. (missing full stop after L and space).

Answer:  Inserted. Thank you.

Question reviewer 6:  Lines 94-95. The sentence can be deleted and the reference to the figure (Figure 1) can be inserted in the sentence preceding it. 

Answer:  The change has been made. Thank you.

Question reviewer 6:  In Figure 1, the colours of the numerals need to be adjusted. The numerals in the graph are now difficult to see because of the lack of contrast. I suggest that the numerals be in black. 

Answer:  The images have been duly altered in accordance with the reviewer's suggestions

Question reviewer 6:  Figure 2. In the caption, descriptions of all parts of the figure should be added and the country, province or other geographic attributes should be indicated.  A mere reference in the legend is not sufficient.  The same applies to Figure 3.

Answer:  The images have been duly altered in accordance with the reviewer's suggestions

Question reviewer 6:  Figure 4. No indication of what is on the Y-axis. It is necessary to add an entry.

Answer:  The images have been duly altered in accordance with the reviewer's suggestions 

Question reviewer 6:  Figure 5. Figure caption must be updated (see note 7). The entries in the figure that are not place names should be translated into English.

Answer:  The images have been duly altered in accordance with the reviewer's suggestions

Question reviewer 6:  Line 232. Remove unnecessary text.

Answer:  Okay.  Corrected.

Question reviewer 6:  Line 285. The reference should not be underlined. 

Okay. Corrected.

Round 2

Reviewer 1 Report

The revised manuscript still has little in science, I cannot guarantee its acceptance by any academic journal, which means, it’s more suitable for a non-academic magazine.

Author Response

Thank you for your comment. Measuring and monitoring sustainability brings challenges of new methods and outcomes; we address a sustainability proposal related to biodiversity, tradition and traditional medicine, all of which are difficult to measure by numbers. As in situ germplasm conservation is an urgent issue and decisions are constantly undertaken, it can be a priority to discuss approaches even if precise measurements are not developed. As tentative methodology, results and discussion are being developed for the development of a scientific approach, we offer it for a publication that is intended for this audience, not as a scientific article but as a communication.

Reviewer 2 Report

The manuscript titled "Gossypium barbadense preserved in situ in Central Brazil by fabric artisans and for medicinal use" is good because it documents the importance of wild plant species “Gossypium barbadense L.” used in Brazil's economic activities. It was written in a well-simplified way, and according to the complex results, it has resulted from surveys and reliance on some historical information. 

Author Response

Thank you for the compliments.

Reviewer 3 Report

After my careful review of this revised version (sustainability-2147866) of the manuscript, titled "Gossypium barbadense preserved in situ in Central Brazil by fabric artisans and for medicinal use". The authors have made a considerable effort to improve and clarify the manuscript. I am satisfied with the corrections made by the authors, and I recommend this paper be accepted.

Author Response

Thank you for your attention and for the compliments.

Reviewer 5 Report

Dear Authors

Thank you for your appropriates responses to review comments.

Regards

Author Response

Thank you for the very important amendments to the manuscript which helped us to improve it.

Reviewer 6 Report

The article has been substantially revised after the first round of reviews, and most of the previous shortcomings have been addressed. Having read the latest version of the article, I would like to make a few minor comments. 

1. Keywords should be in alphabetical order.

2. Lines 56-57. The sentence needs to be edited to make it more fluent.

3. The captions of Figure 2, Figure 3 and Figure 5 should be clarified to make it clear what is being presented. For example, does Figure 2 only show the cultivation sites or does it relate to the study (research sites)? What was collected at the points indicated in Figure 3? Generic name must be unabbreviated in Figure 5.

4. Figure 4 needs to be corrected. The image and entries are blurred. What do the authors mean by "subject age?" The font of the entry needs to be corrected.

5. There are technical problems in the text (formatting, missing or too many spaces, etc.). There are occasional language style problems.

Author Response

Thank you for your attention and suggestions. Keywords are now in alphabetical order. Lines 56-57 were modified: ‘Social and environmental changes have put in strong threat the in situ conservation of germplasm, and the tools to describe, measure, and encourage its continued maintenance are scarce. Considering the hypothesis that it is possible to characterize social and economic aspects leadingto in situ conservation, the objectives of the expeditions and interviews were to relate G. barbadense maintenance to community sustainability.’ We collected seeds for ex situ maintenance, using negative temperatures, at Embrapa germplasm bank, at Brasilia . We added this information in materials and methods. Legends of all figures were corrected. The text has been reviewed and corrected.

Round 3

Reviewer 1 Report

I said many times, the manuscript has no science and cannot be accepted by an academic journal.

Author Response

The question of how sustainability generate questions that new scientific approaches and, furthermore, how to proceed to ensure they consists in real and valuable science, is more complex than can be stated in one paragraph. Therefore, briefly, sustainability often reflects to local, regional, specific aspects, and in order to be called science, the approach must in essence be reproducible elsewhere, and generate data that can be repeated by other authors. To relate sustainability with in situ maintenance of genetic resources and its value, we use interviews, morphological characterization of genetic diversity, and public government data, and we believe it may be applicable in any regions.

Reviewer 6 Report

Most of the previous shortcomings have been addressed in the current version of the article. A few minor remarks: 

1. Figure 4. The font of the entries is too small and difficult to read, especially the y-axis value. 

2. Line 166. What is the timeframe for such income? 

3. Lines 212-2016. The formatting of the paragraph needs to be adjusted.

4. Reference [21] without title and other attributes. To be completed.

Author Response

  1. Figure 4. The font of the entries is too small and difficult to read, especially the y-axis value. 

Response: Thanks, it was improved

  1. Line 166. What is the timeframe for such income?

Response: One year. Thanks a lot for the extremely relevant observation, this information was added.

  1. Lines 212-2016. The formatting of the paragraph needs to be adjusted.

Response: It was adjusted.

  1. Reference [21] without title and other attributes. To be completed.

Response: It was completed.
